# Oracle-Preserving Latent Flows

**Alexander Roman** [ID], **Roy T. Forestano** [†][ID], **Konstantin T. Matchev** [*,†][ID], **Katia Matcheva** [†][ID] **and Eyup B. Unlu** [†][ID]

Institute for Fundamental Theory, Physics Department, University of Florida, Gainesville, FL 32611, USA; alexroman@ufl.edu (A.R.); roy.forestano@ufl.edu (R.T.F.); katia@phys.ufl.edu (K.M.); eyup.unlu@ufl.edu (E.B.U.)
* Correspondence: matchev@ufl.edu
† These authors contributed equally to this work.

**Abstract:** A fundamental task in data science is the discovery, description, and identification of any symmetries present in the data. We developed a deep learning methodology for the simultaneous discovery of multiple non-trivial continuous symmetries across an entire labeled dataset. The symmetry transformations and the corresponding generators are modeled with fully connected neural networks trained with a specially constructed loss function, ensuring the desired symmetry properties. The two new elements in this work are the use of a reduced-dimensionality latent space and the generalization to invariant transformations with respect to high-dimensional oracles. The method is demonstrated with several examples on the MNIST digit dataset, where the oracle is provided by the 10-dimensional vector of logits of a trained classifier. We find classes of symmetries that transform each image from the dataset into new synthetic images while conserving the values of the logits. We illustrate these transformations as lines of equal probability ("flows") in the reduced latent space. These results show that symmetries in the data can be successfully searched for and identified as interpretable non-trivial transformations in the equivalent latent space.

**Keywords:** symmetry transformations; symmetry invariant; Lie groups; Lie algebras; supervised learning; deep learning; encoder–decoder; MNIST handwritten digit dataset

## 1. Introduction

Symmetries permeate the world around us and can be found at all scales—from the microscopic description of subatomic particles in the standard model (SM) to the large-scale structure of the universe. Symmetries are used as a guiding principle in contemporary science [1], as well as in our everyday interactions. Therefore, a fundamental task in data science is the discovery, description, and identification of the symmetries present in a given dataset. According to Noether's theorem [2], the presence of a continuous symmetry in data implies that there exists a conservation law that is universally applicable and indispensable in understanding the system's behavior and evolution. At the same time, symmetries can be perceived as aesthetically pleasing in the arts and be used to recognize and evaluate the work of schools and individual artists [3]. In both theoretical physics and abstract mathematics, there is a rich tradition of studying symmetries and their underlying group properties, which prove interesting in their own right [4].

Applications of machine learning (ML) to the study of symmetries have been pursued by a number of groups in various contexts. Previous research has concentrated on examining the correlation between a specific symmetry and a learned representation of the data. This has been explored in various domains, such as astronomy [5] and particle physics [6]. Similarly, investigations have been conducted into how a symmetry can be incorporated within the ML architecture, e.g., in the embedding layer of a neural network (NN) [7]. In addition, alternative approaches have proposed for the development of specialized ML architectures, such as equivariant NNs, which are intentionally designed with specific symmetry properties from the beginning [8]. For example, Lorentz symmetry can be usefully enforced in particle physics to classify jets [9,10], to identify top quark decays [11,12],

or for anomaly detection [13]. Gauge symmetry can be similarly implemented in lattice simulations [14] (for a review, see [15]). When symmetries are incorporated directly into the ML model, it becomes more economical (in terms of learned representations), interpretable, and trainable. The explorations in this field can also be expanded to include discrete symmetries, particularly permutation symmetries [16,17]. These endeavors lay the groundwork for data-driven investigations in search of new physics, whereby the data are rigorously tested for any potential deviations from the well-established symmetries of the standard model [18–20].

Machine learning techniques have recently been applied to more formal mathematical questions that traditionally have fallen within the domain of theorists, e.g., performing symbolic computations [21,22] or deriving analytical formulas by training a symbolic regression on synthetic data [23–28]. The benefits of symbolic deep learning have been demonstrated in a number of usage cases, including astrophysics [29–31], astronomy [5,32], exoplanets [33], particle physics [34–36], material [37], and social science [38]. A comprehensive understanding of the symmetries inherent in a problem often leads to the identification of conserved quantities [39,40] or provides insights into a more fundamental description of the problem [41]. Machine learning has been employed in various applications, including the detection of symmetries in potentials [3,7,42], determining the symmetry relationship between input pairs [43], exploring scale-invariant and conformal symmetries [44], and the landscape of string theory [45–47]. Recent advancements have leveraged generative adversarial networks (GANs) to learn transformations that preserve probability distributions [48]. Furthermore, ML applications have extended into the realm of group theory, which serves as the mathematical language for abstract symmetries. For instance, ML has been utilized to study irreducible representations of Lie groups [49], as well as to obtain the Lie group generators that reflect symmetries present in the data [50–53].

The primary objective of this paper is to develop a deep learning approach that emulates the cognitive processes of traditional theorists and has the ability to identify and classify the complete set of (continuous) symmetries present in a given dataset, all "from first principles". This means that the method does not rely on any preconceived assumptions or biases. The only inputs required for our methodology are a labeled dataset $(\mathbf{x}; y)$ similar to the one shown in Equation (1) and a vector oracle $(\vec{\varphi}(\mathbf{x}))$, which can either be learned from the dataset itself or provided externally. Our study complements or extends previous related works [3,7,42,50–53]. The procedure is general and does not require a priori knowledge of what potential symmetries might be present in the dataset. Instead, the symmetries are learned from scratch. A public code for this paper is available online [54].

## 2. Definition of the Problem

Our starting point is a labeled dataset containing $m$ samples of $n$ features and $k$ targets:

$$
\begin{array}{cccccccc}
x_1^{(1)}, & x_1^{(2)}, & \ldots, & x_1^{(n)}; & y_1^{(1)}, & y_1^{(2)}, & \ldots, & y_1^{(k)} \\
x_2^{(1)}, & x_2^{(2)}, & \ldots, & x_2^{(n)}; & y_2^{(1)}, & y_2^{(2)}, & \ldots, & y_2^{(k)} \\
\vdots & \vdots & \vdots & \vdots & \vdots & \vdots & \vdots & \vdots \\
x_m^{(1)}, & x_m^{(2)}, & \ldots, & x_m^{(n)}; & y_m^{(1)}, & y_m^{(2)}, & \ldots, & y_m^{(k)}
\end{array}
\tag{1}
$$

We use boldface vector notation

$$
\mathbf{x} \equiv \{x^{(1)}, x^{(2)}, \ldots, x^{(n)}\} \in \mathbb{R}^n
\tag{2}
$$

for the $n$-dimensional input features and arrow vector notation for the $k$-dimensional target vectors.

$$
\vec{y} \equiv \{y^{(1)}, y^{(2)}, \ldots, y^{(k)}\} \in \mathbb{R}^k.
\tag{3}
$$

The dataset (1) can then be written in a compact form as $\{\mathbf{x}_i; \vec{y}_i\}$, where $i = 1, 2, \ldots, m$.

In order to define a symmetry of the data (1), we utilize the vector function $\vec{\varphi}(\mathbf{x})$, which plays the role of an oracle producing the corresponding target labels $(\vec{y}_1, \vec{y}_2, \ldots, \vec{y}_m)$:

$$\vec{y}_i \equiv \vec{\varphi}(\mathbf{x}_i), \qquad i = 1, 2, \ldots, m. \tag{4}$$

This is an ideal classifier on the given dataset. In data science applications (our primary interest in this paper), the vector oracle ($\vec{\varphi} : \mathbb{R}^n \to \mathbb{R}^k$) needs to be learned numerically from the dataset (1) via standard regression methods. On the other hand, in certain more formal theoretical applications, the oracle ($\vec{\varphi}$) can already be provided to us externally in the form of an analytical function. The methodology developed in this paper applies to both of these situations.

With this setup, the main goal is to derive a symmetry transformation ($\mathbf{f} : \mathbb{R}^n \to \mathbb{R}^n$)

$$\mathbf{x}' = \mathbf{f}(\mathbf{x}), \tag{5}$$

which preserves the $\vec{\varphi}$-induced labels (4) of our dataset (1). In other words, we want to find the function ($\mathbf{f}(\mathbf{x})$) for which

$$\vec{\varphi}(\mathbf{x}'_i) \equiv \vec{\varphi}(\mathbf{f}(\mathbf{x}_i)) = \vec{\varphi}(\mathbf{x}_i), \quad \forall i = 1, 2, \ldots, m. \tag{6}$$

In general, a given dataset exhibits several different symmetries, which can now be examined and categorized from the point of view of group theory. For this purpose, one needs to focus on infinitesimal symmetry transformations and study the corresponding set of distinct generators ($\{\mathbf{J}_\alpha\}, \alpha = 1, 2, \ldots, N_g$). A given set of generators ($\{\mathbf{J}_\alpha\}$) forms a Lie algebra if the closure condition is satisfied, i.e., if all Lie brackets $\left[ \cdot, \cdot \right]$ can be represented as linear combinations of the generators already present in the set:

$$\left[\mathbf{J}_\alpha, \mathbf{J}_\beta\right] = \sum_{\gamma=1}^{N_g} a_{[\alpha\beta]\gamma} \mathbf{J}_\gamma. \tag{7}$$

The coefficients ($a_{[\alpha\beta]\gamma}$) are the structure constants of the symmetry group present in our dataset (the square bracket index notation reminds the reader that they are antisymmetric in their first two indices: $a_{[\alpha\beta]\gamma} = -a_{[\beta\alpha]\gamma}$). The simplest algebras (referred to as Abelian) are those for which all generators commute and, hence, have vanishing structure constants.

## 3. Method

Following [52,53], we model the function $\mathbf{f}$ with a neural network ($\mathcal{F}_{\mathcal{W}}$) with $n$ neurons in the input and output layers, corresponding to the $n$ transformed features of the data point $\mathbf{x}'$. The trainable network parameters (weights and biases) are generically denoted by $\mathcal{W}$. During training, they evolve and converge to the corresponding *trained* values ($\widehat{\mathcal{W}}$) of the parameters of the trained network ($\mathcal{F}_{\widehat{\mathcal{W}}}$), i.e., the hat symbol denotes the result of the training. In order to ensure the desired properties of the network, we design a loss function with the following elements.

**Invariance:** In order to enforce invariance under transformation (5), we include the following mean squared error (MSE) term in the loss function $L$:

$$L_{\text{inv}}(\mathcal{W}, \{\mathbf{x}_i; \vec{y}_i\}) = \frac{1}{m} \sum_{i=1}^{m} [\vec{\varphi}(\mathcal{F}_{\mathcal{W}}(\mathbf{x}_i)) - \vec{y}_i]^2. \tag{8}$$

A NN trained with this loss function produces an arbitrarily general (finite) symmetry transformation ($\mathcal{F}_{\widehat{\mathcal{W}}}$) parameterized by the values of the trained network parameters ($\widehat{\mathcal{W}}$). The particular instantiation of $\mathcal{F}_{\widehat{\mathcal{W}}}$ depends on the initialization of the network parameters, so by repeating the procedure with different initializations, one obtains a family of symmetry transformations.

**Infinitesimality:** In order to focus on the symmetry *generators*, we restrict ourselves to infinitesimal transformations ($\delta \mathcal{F}$) in the vicinity of the identity transformation ($\mathbf{I}$):

$$\delta \mathcal{F} \;\equiv\; \mathbf{I} + \varepsilon\, \mathcal{G}_{\mathcal{W}}\,, \tag{9}$$

where $\varepsilon$ is an infinitesimal parameter, and the parameters ($\mathcal{W}$) of the new neural network ($\mathcal{G}$) are forced to be finite. The loss function (8) can then be rewritten as

$$L_{\mathrm{inf}}(\mathcal{W}, \{\mathbf{x}_i; \vec{y}_i\}) = \frac{1}{m\varepsilon^2} \sum_{i=1}^{m} [\vec{\varphi}(\mathbf{x}_i + \varepsilon \mathcal{G}_{\mathcal{W}}(\mathbf{x}_i)) - \vec{y}_i]^2\,, \tag{10}$$

with an extra factor of $\varepsilon^2$ in the denominator to compensate for the fact that generic transformations scale as $\varepsilon$ [42]. In addition, we add a normalization loss term:

$$L_{\mathrm{norm}}(\mathcal{W}, \{\mathbf{x}_i\}) = \frac{1}{m} \sum_{i=1}^{m} [\|\mathcal{G}_{\mathcal{W}}(\mathbf{x}_i)\| - 1]^2 + \frac{1}{m} \sum_{i=1}^{m} \left[\|\mathcal{G}_{\mathcal{W}}(\mathbf{x}_i)\| - \overline{\|\mathcal{G}_{\mathcal{W}}(\mathbf{x}_i)\|}\right]^2\,, \tag{11}$$

where the overline in the last term indicates sample averaging.

After minimization of the loss function, the trained NN ($\mathcal{G}_{\widehat{\mathcal{W}}}$) represents a corresponding generator

$$\mathbf{J} \;=\; \mathcal{G}_{\widehat{\mathcal{W}}}\,, \tag{12}$$

where

$$\widehat{\mathcal{W}} \equiv \arg\min_{\mathcal{W}} \left( L_{\mathrm{inf}} + h_{\mathrm{norm}} L_{\mathrm{norm}} \right) \tag{13}$$

are the learned values of the NN parameters ($h_{\mathrm{norm}}$ is a hyperparameter usually set to 1). By repeating the training $N_g$ times under different initial conditions ($\mathcal{W}_0$), one obtains a set of $N_g$ (generally distinct) generators ($\{\mathbf{J}_\alpha\}$, $\alpha = 1, 2, \ldots, N_g$).

**Orthogonality:** To ensure that the generators ($\{\mathbf{J}_\alpha\}$) are distinct, we introduce an additional orthogonality term to the loss function.

$$L_{\mathrm{ortho}}(\mathcal{W}, \{\mathbf{x}_i\}) = \frac{1}{m} \sum_{i=1}^{m} \sum_{\alpha < \beta}^{N_g} \left[ \mathcal{G}_{\mathcal{W}_\alpha}(\mathbf{x}_i) \cdot \mathcal{G}_{\mathcal{W}_\beta}(\mathbf{x}_i) \right]^2\,. \tag{14}$$

**Group structure:** In order to test whether a certain set of *distinct* generators ($\{\mathbf{J}_\alpha\}$) found in the previous steps generates a group, we need to check the closure of the algebra (7), e.g., by minimizing

$$L_{\mathrm{closure}}(a_{[\alpha\beta]\gamma}) = \sum_{\alpha < \beta} \mathrm{Tr}\left( \mathbf{C}_{[\alpha\beta]}^{T} \mathbf{C}_{[\alpha\beta]} \right), \tag{15}$$

with respect to the candidate structure constant parameters ($a_{[\alpha\beta]\gamma}$), where the closure mismatch is defined by

$$\mathbf{C}_{[\alpha\beta]}(a_{[\alpha\beta]\gamma}) \equiv \left[\mathbf{J}_\alpha, \mathbf{J}_\beta\right] - \sum_{\gamma=1}^{N_g} a_{[\alpha\beta]\gamma} \mathbf{J}_\gamma. \tag{16}$$

Since $L_{\mathrm{closure}}$ is positive and semidefinite, $L_{\mathrm{closure}} = 0$ indicates that the algebra is closed and that we are therefore dealing with a genuine (sub)group.

In principle, the number of generators ($N_g$) is a hyperparameter that must be specified ahead of time. Therefore, when a closed algebra for a given $N_g$ value is found, it is only guaranteed to be a subalgebra of the full symmetry group, and one must proceed to also test higher values for $N_g$. The full algebra then corresponds to the maximum value of $N_g$ for which a closed algebra of distinct generators is found to exist [52,53].

## 4. Simulation Setup

In order to illustrate the method, we use the standard MNIST dataset [55] consisting of 60,000 images of 28 by 28 pixels. In other words, the dataset (1) consists of $m = 60,000$ samples of $n = 28 \times 28 = 784$ features. The images are labeled as digits from 0 to 9. The classic classification task for this dataset is to build an oracle $(\vec{y} = \vec{\varphi}(\mathbf{x}))$ that calculates a ten-dimensional vector of 'logits' in the output layer (this raw output is typically normalized into respective probabilities with a softmax function). An image is then classified as $\arg\max(\vec{\varphi}(\mathbf{x}))$.

When applied to the MNIST dataset, the general problem presented in Section 2 can be formulated as follows: what types of transformations (5) can be performed on *all of* the original images in the dataset so that the 10-dimensional logit vector $(\vec{\varphi}(\mathbf{x}))$ is conserved, i.e., the symmetry Equation (6) is satisfied? Note that conserving all ten components of the oracle function as in Equation (6) is a much stronger requirement than simply demanding that the prediction for each image remains the same. For the latter, it is sufficient to ensure that the map (**f**) is such that

$$\arg\max(\vec{\varphi}(\mathbf{f}(\mathbf{x}))) = \arg\max(\vec{\varphi}(\mathbf{x})). \tag{17}$$

As discussed in Section 3 and illustrated with the examples in the next section, we apply deep learning to derive the desired transformations (**f**). The neural network $(\mathcal{F}_\mathcal{W})$ is implemented as a sequential feed-forward neural network in PYTORCH [56]. Optimizations are performed with the ADAM optimizer with a learning rate between $3 \times 10^{-5}$ and 0.03. The loss functions were designed to achieve a fast and efficient training process without the need for extensive hyperparameter tuning. The training and testing data (1) were obtained from the standard MNIST dataset [55].

### 4.1. Trivial Symmetries from Ignorable Features

Before proceeding, let us develop some intuition by discussing the trivial symmetries that are present in the dataset in analogy to the treatment of cyclic (also called ignorable) coordinates in classical mechanics [57]. In classical mechanics, if the Lagrangian of a system does not contain a certain coordinate, then that coordinate is said to be cyclic or ignorable, and its corresponding conjugate momentum is conserved. Let us ask whether there are any such ignorable features in our case. The two heat maps in Figure 1 show the maximum value for each pixel (left panel), as well as the mean value for each pixel (right panel), when averaged over the whole dataset. Figure 1 reveals that there are a number of pixels near the corners and the edges of the image that contain no data at all. This suggests that a robust classifier is insensitive to the values of those pixels, i.e., these pixels behave similarly to the ignorable coordinates seen in classical mechanics. This observation is further evidenced by the fact that the highest, i.e., least informative, principal components across the features (not shown here) are linear combinations of those corner and/or edge pixels. We are not interested in such trivial symmetries.

It may be possible to find non-trivial maps directly between images using a deep NN, but training is predicated on the existence of a metric capable of capturing non-trivial structure in the dataset. (For example, rotating an image slightly may result in a symmetry with respect to a classifier, but it changes the values of many pixels. Hence, the value of a naive metric such as the MSE between these two images would be large. On the other hand, the distance as measured by a more sophisticated metric, such as the Earth Mover's Distance, would be small.) One candidate for such a metric is the Earth Mover's Distance, which solves an optimal transport problem between two distributions. However, it is too computationally expensive to be used in the loss function. In what follows, we first reduce the dimensionality of the dataset and explore symmetry transformations in the corresponding $\ell$-dimensional latent space $(\mathbb{R}^\ell)$ with $\ell \ll n$ because in latent space, the Euclidean metric captures rich structure.

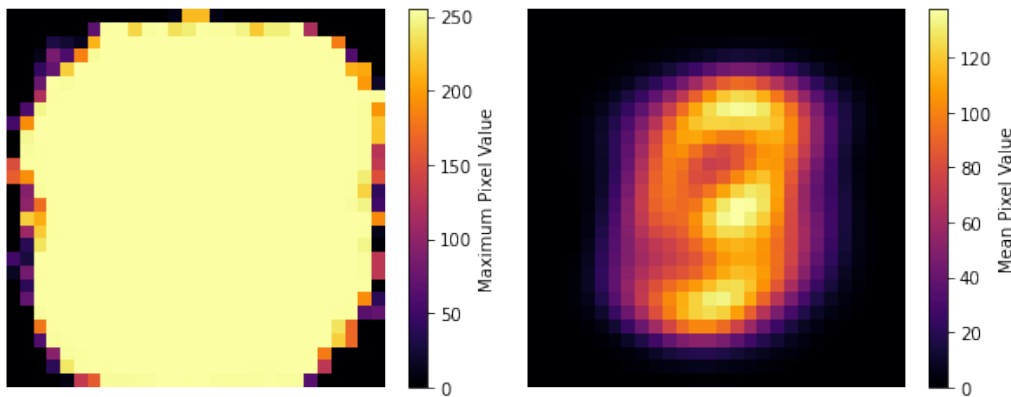

**Figure 1.** Heat maps of the maximum value (**left** panel) and the mean value (**right** panel) of each pixel in the MNIST dataset. The individual pixel values in the dataset range from 0 to 255.

### 4.2. Dimensionality Reduction

As shown in Figure 2, instead of looking for transformations (**f**) in the original feature space ($\mathbb{R}^n$) that are symmetries that preserve the oracle ($\vec{\varphi}$) from (4), we choose to search for transformations (**g**) acting in a latent space ($\mathbb{R}^\ell$), which are symmetries that preserve the *induced* oracle ($\vec{\psi} : \mathbb{R}^\ell \to \mathbb{R}^k$). For this purpose, we train an autoencoder, the architecture of which is shown in Figure 3, consisting of an encoder ($\mathbf{E} : \mathbb{R}^n \to \mathbb{R}^\ell$) and a decoder ($\mathbf{D} : \mathbb{R}^\ell \to \mathbb{R}^n$). The latent vectors ($\mathbf{z}_i \in \mathbb{R}^\ell$) are defined as

$$\mathbf{z}_i \;\equiv\; \mathbf{E}(\mathbf{x}_i), \qquad i = 1, 2, \ldots, m. \tag{18}$$

The induced classifier ($\vec{\psi} : \mathbb{R}^\ell \to \mathbb{R}^k$) is trained as

$$\vec{y}_i = \vec{\psi}(\mathbf{z}_i) = \vec{\psi}(\mathbf{E}(\mathbf{x}_i)), \quad i = 1, 2, \ldots, m; \tag{19}$$

built as a fully connected dense NN with three hidden layers of sizes 128, 128, and 32; and trained with the categorical cross-entropy loss.

With those tools in hand, we look for symmetry transformations (**g**) $: \mathbb{R}^\ell \to \mathbb{R}^\ell$ in the latent space in analogy to Equation (9):

$$\mathbf{z}' \;=\; \mathbf{z} + \varepsilon\, \mathbf{g}(\mathbf{z}), \tag{20}$$

which preserves the new oracle ($\psi$) in analogy to (6)

$$\vec{\psi}(\mathbf{z}'_i) \;\equiv\; \vec{\psi}(\mathbf{z} + \varepsilon\, \mathbf{g}(\mathbf{z})) = \vec{\psi}(\mathbf{z}_i), \quad \forall i = 1, 2, \ldots, m. \tag{21}$$

A single transformation (**g**) is represented by a fully connected dense NN with various architectures, as necessitated by the complexity of the exercise, and trained with the loss functions described in Section 3. Once such symmetry transformation (**g**) in the latent space is found, and its effect on the actual images can be illustrated and analyzed with the help of the decoder (**D**).

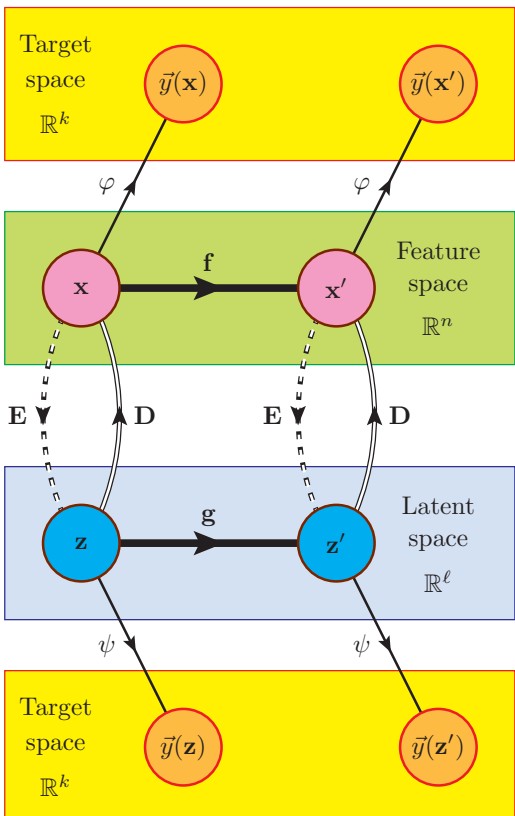

**Figure 2.** Flow chart of the different data processing steps discussed in this paper: an encoder ($\mathbf{E} : \mathbb{R}^n \to \mathbb{R}^\ell$), a decoder ($\mathbf{D} : \mathbb{R}^\ell \to \mathbb{R}^n$), a transformation in the feature space ($\mathbf{f} : \mathbb{R}^n \to \mathbb{R}^n$), a transformation in the latent space ($\mathbf{g} : \mathbb{R}^\ell \to \mathbb{R}^\ell$), a trained vector oracle ($\vec{\varphi} : \mathbb{R}^n \to \mathbb{R}^k$), and a trained vector oracle ($\vec{\psi} : \mathbb{R}^\ell \to \mathbb{R}^k$). The transformations ($\mathbf{f}$) and $\mathbf{g}$ are symmetries if $\vec{y}(\mathbf{x}') = \vec{y}(\mathbf{x})$ and $\vec{y}(\mathbf{z}') = \vec{y}(\mathbf{z})$, respectively.

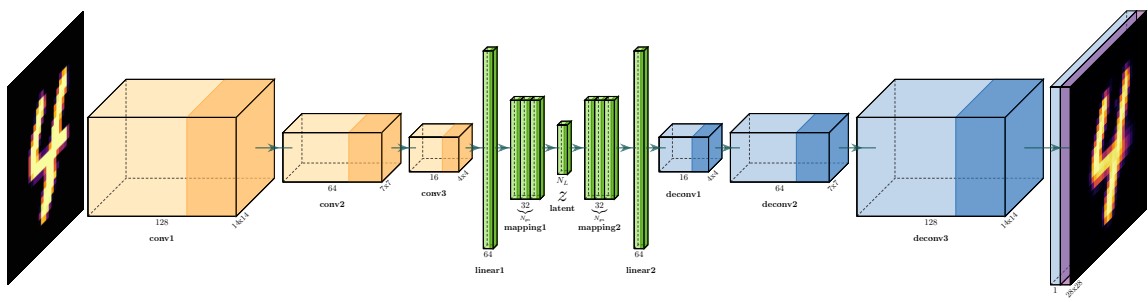

**Figure 3.** The network architecture of our autoencoder consisting of an encoder ($\mathbf{E}$) and a decoder ($\mathbf{D}$). The yellow modules are convolutional layers, the green modules are fully connected layers, and the blue modules are convolution transpose layers. The dark shaded regions indicate ReLU activation functions.

## 5. Examples

For the main exercise presented in Section 5.3 below, we use the complete set of digits and a 16-dimensional latent space in which the results are difficult to plot and visualize, which is why we begin with a couple of toy examples for which we consider only two classes—the zeros and the ones—and either a two-dimensional latent space (Section 5.1) or a three-dimensional latent space (Section 5.2).

### 5.1. Two Categories and $\ell = 2$ Latent Variables

In this subsection, we perform a toy binary classification exercise in $\ell = 2$ latent space dimensions. We keep only the images of class "0" and "1", which are then randomly train–test-split in a ratio of 3:1. The training set is used to train the autoencoder shown in Figure 3 and the classifier ($\psi$). Since this example is a simple binary classification, $\psi$ is set to have a single output layer neuron, the logit (raw output) of which is fed into a sigmoid function. The results are shown in Figure 4 in the plane of the two latent-space variables $(z^{(1)}, z^{(2)})$. The red and blue points denote the set of validation images with true labels 0 and 1, respectively. The white star symbols mark the centers of these two clusters, which we refer to as "Platonic" (in the sense that they are the ideal representatives of their respective classes) images of the digits zero and one. By moving along the straight, white dashed line between them, we pass through points in the latent space, which (after decoding) produce images that smoothly interpolate between the platonic zero and the platonic one. This variation is illustrated in the top panel of Figure 5. Motion along the white dashed line is therefore not a symmetry transformation, since it changes the meaning of the image. It is the motion in the orthogonal direction that we are interested in, since that is the direction in which, while the actual image is changing, its interpretation by the classifier is not.

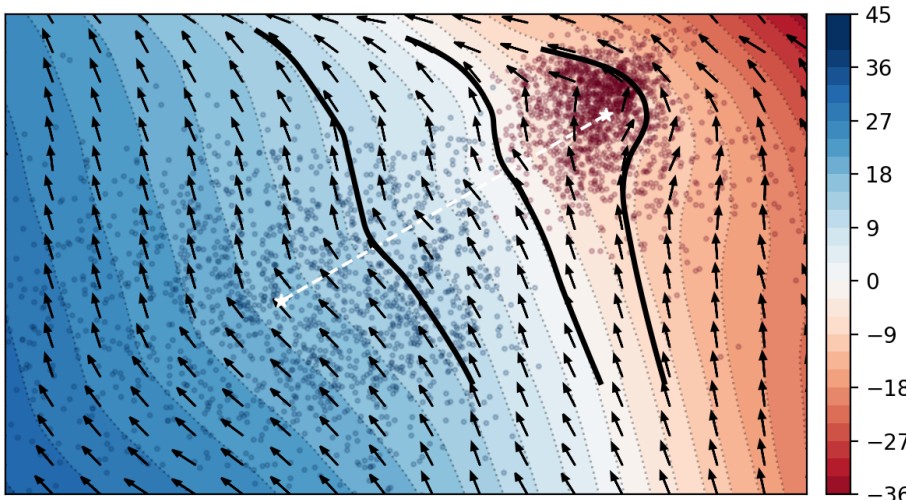

**Figure 4.** The results of the exercise described in Section 5.1 illustrated in two-dimensional latent space. The red and blue points represent validation images with true labels of 0 and 1, respectively, and the white stars (connected by a straight dashed line) denote the centers of these two clusters. The heat map shows the unscaled output from the single neuron in the output layer of the $\psi$ classifier. The superimposed vector field visualizes the symmetry transformation (**g**) found by our method. The black solid lines are three representative symmetry streamlines used for the illustrations in Figure 5.

Once we train the neural network for **g**, we obtain a vector field in the latent space. The *flow* of this vector field is illustrated by black arrows in Figure 4. The three black solid lines are three representative symmetry streamlines used for the illustrations in the bottom three panels in Figure 5. The leftmost streamline passes near the Platonic streamline and therefore represents a series of images that are interpreted by the classifier ($\psi$) as "ones" with very high probability (see the blue line in the second panel of Figure 5). The corresponding row of decoded images in the second panel reveals that the symmetry transformation has the effect of rotating the digit "one" counterclockwise.

The rightmost streamline in Figure 4, on the other hand, passes through the region near the Platonic "zero" and therefore contains images that are almost surely interpreted as zeros by the classifier, as evidenced by the red line in the last panel of Figure 5. The corresponding row of decoded images in the last panel reveals that the symmetry transformation has

the effect of not only rotating the digit "zero" counterclockwise but also simultaneously stretching and enlarging the image.

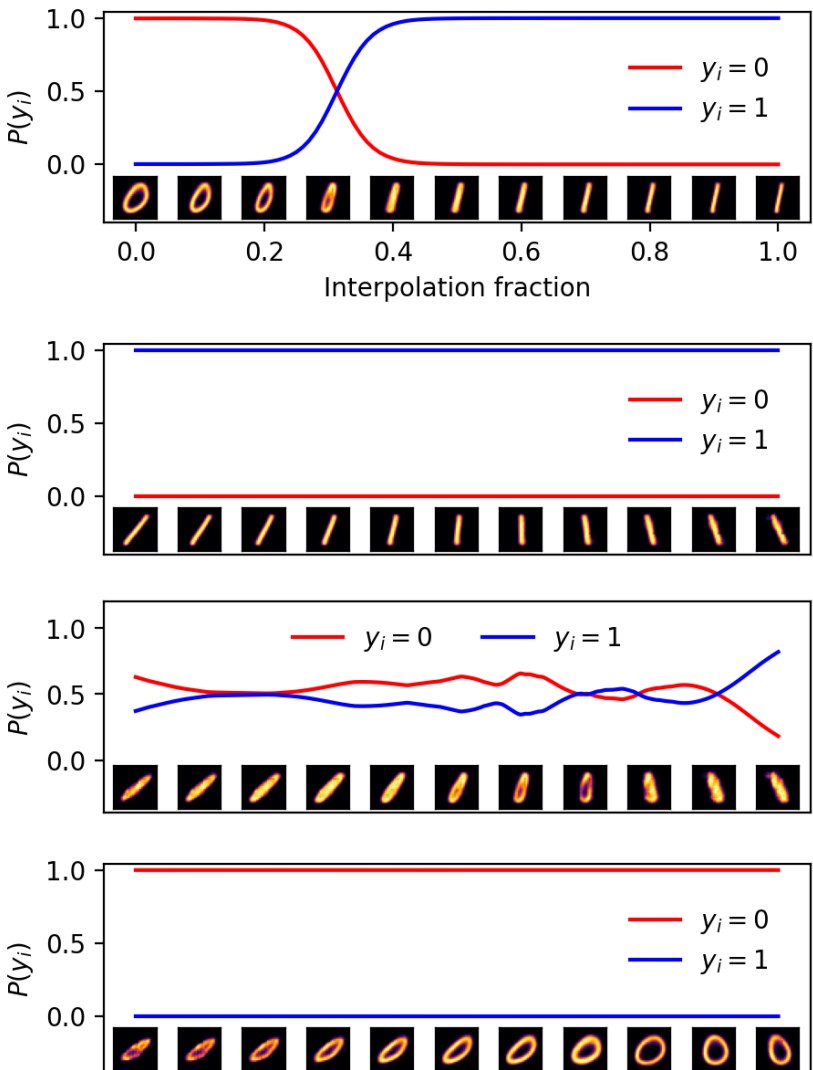

**Figure 5.** Explorations of the latent space from Figure 4 by following the white dashed line (top panel), the left solid black line (second panel), the middle solid black line (third panel), or the right solid black line (fourth panel). Each panel shows the likelihood of an image being zero (red line) or one (blue line) along the respective latent-space trajectory. At the base of each panel, we show a row of representative images after applying the decoder ($\mathbf{D}(\mathbf{z})$).

The middle streamline in Figure 4 passes through the boundary region between the two category clusters and therefore generates images that are rather inconclusive according to the classifier. This is confirmed in the third panel of Figure 5, which shows that the likelihoods of a "zero" and "one" are comparable along that streamline (It is noteworthy that the line is somewhat unstable because there are very few samples in this region; hence, the generator model is less reliable here). The decoded images are confusing to interpret, even for a human and are rotated counterclockwise in a similar fashion to the images in the other panels.

An important feature of our scheme is the ability to find multiple non-trivial symmetries and vice-versa in order to determine when no additional symmetries are possible. Since we chose a compressed representation with only two dimensions, one of which is constrained by the oracle, we are left with a single symmetry degree of freedom. Therefore, if we attempt to identify a second symmetry (by simultaneously training two NNs for $\mathbf{g}$,

as explained in Section 3), the training converges to a large loss value, implying that all desired conditions cannot be satisfied simultaneously. Specifically, invariance requires that the flows follow the contours of equal likelihood, forcing them to align, which contradicts the orthogonality condition. This dilemma is illustrated in Figure 6, where we repeat the previous exercise for the case of $N_g = 2$. The two latent flows represented by the black and blue arrows, respectively, represent an attempt to strike an optimal balance between these competing and irreconcilable requirements.

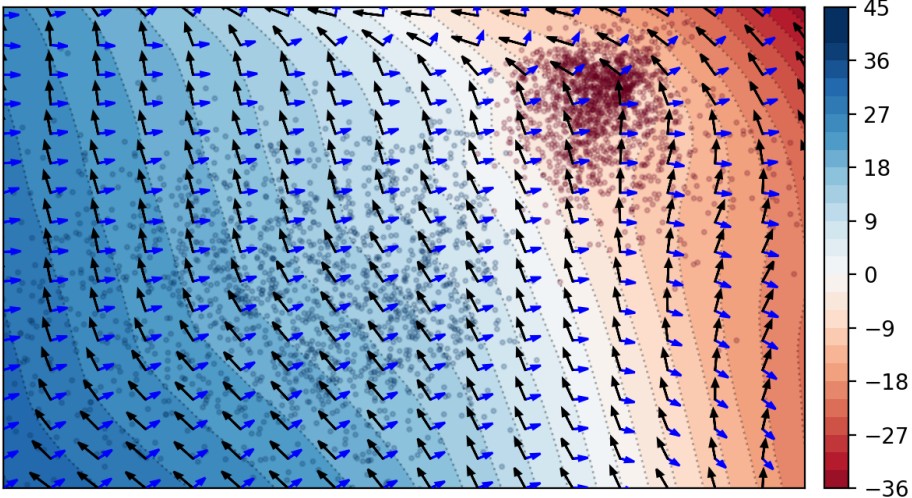

**Figure 6.** As in Figure 4 but requiring two separate orthogonal symmetry transformations ($\mathbf{g}_1$ (black arrows) and $\mathbf{g}_2$ (blue arrows)).

### 5.2. Two Categories and $\ell = 3$ Latent Variables

In this subsection, we expand the latent space of our toy example to $\ell = 3$ dimensions, which, in contrast to the example presented in the previous subsection, allows us to find a second, non-trivial, latent flow that generates a symmetry. The analysis proceeds as before, except the autoencoder is retrained with a three-dimensional bottleneck. The final result is presented in Figure 7, which shows the three-dimensional latent space, together with the validation data that form two clusters of zeros and ones. Then, we superimpose three surfaces that are level sets of the oracle ($\psi$) with likelihoods of "zero" of 0.9999, 0.5, and 0.0001, respectively. A generic local symmetry transformation is tangential to these surfaces, which is precisely the result we find when we train a NN for a single symmetry transformation ($\mathbf{g}$).

The real power of our method lies in its ability to simultaneously find multiple orthogonal symmetry generators. When we retrain for the case of $N_g = 2$, we are able to find two orthogonal vector flows, as illustrated in Figure 7 with the yellow and red arrows (for simplicity, we only show a small number of vectors sampled from the top surface). Note that all arrows are tangential to the level-set surface, as expected for an invariant latent flow.

We chose this particular toy example because it represents the most complicated situation in which there are multiple non-trivial generators that can still be easily visualized. Once we generalize to higher dimensions in the next subsection, such a simple visualization is not possible, but the results can be intuitively understood in a similar fashion.

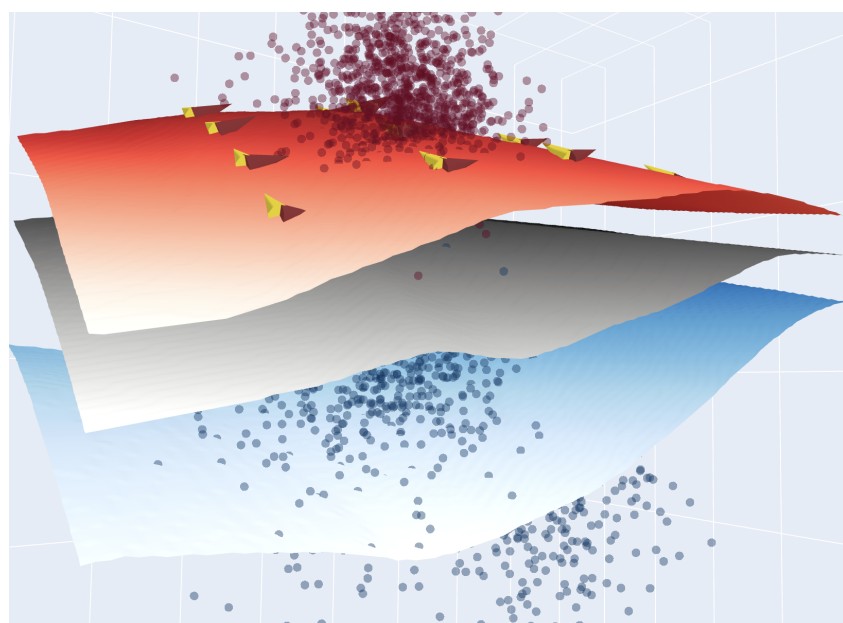

**Figure 7.** An illustration of the three-dimensional latent space of the example in Section 5.2. The red and blue points represent validation images with true labels of 0 and 1, respectively. The three surfaces show three level sets of the oracle ($\psi$) with likelihoods of "zero" of 0.9999, 0.5, and 0.0001, respectively (from top to bottom). The yellow and red arrows denote vectors sampled from the two latent flows found by our method (for simplicity, we only sample points on the upper surface).

### 5.3. Ten Categories and $\ell = 16$ Latent Variables

In this subsection, we present our final example in which we consider all ten classes of digits and use an autoencoder with an $\ell = 16$ dimensional bottleneck. To improve performance, we found it useful to increase the number of mapping layers (i.e., those immediately before and after the bottleneck) from one to three. Proceeding as before, we train a classifier ($\vec{\psi}$) with 10 softmax outputs and a neural network for the symmetry transformation (**g**). The result is illustrated in Figure 8, where, in analogy to Figure 5, we show a series of decoded images along a streamline of the latent flow. The center images in Figure 8 represent the platonic digits in the dataset. From each of those ten starting points, we follow the respective streamline for 6000 steps of $\varepsilon = \pm 10^{-3}$. The three images to the left and to the right of the central one are obtained after 2000, 4000, and 6000 such steps in each direction, respectively. The images in each row are classified correctly with a predicted probability 1.0. This validates our method for the case of a single symmetry flow.

We also simultaneously trained multiple generators and found non-trivial orthogonal flows, even for relatively large numbers of generators ($N_g \sim 10$), which hints at the presence of non-Abelian symmetries. This study only scratches the surface of a promising new research direction to reveal the rich symmetry structure of complex datasets, which we leave for future work. The end goal should be a rigorous understanding of their algebraic and topological properties.

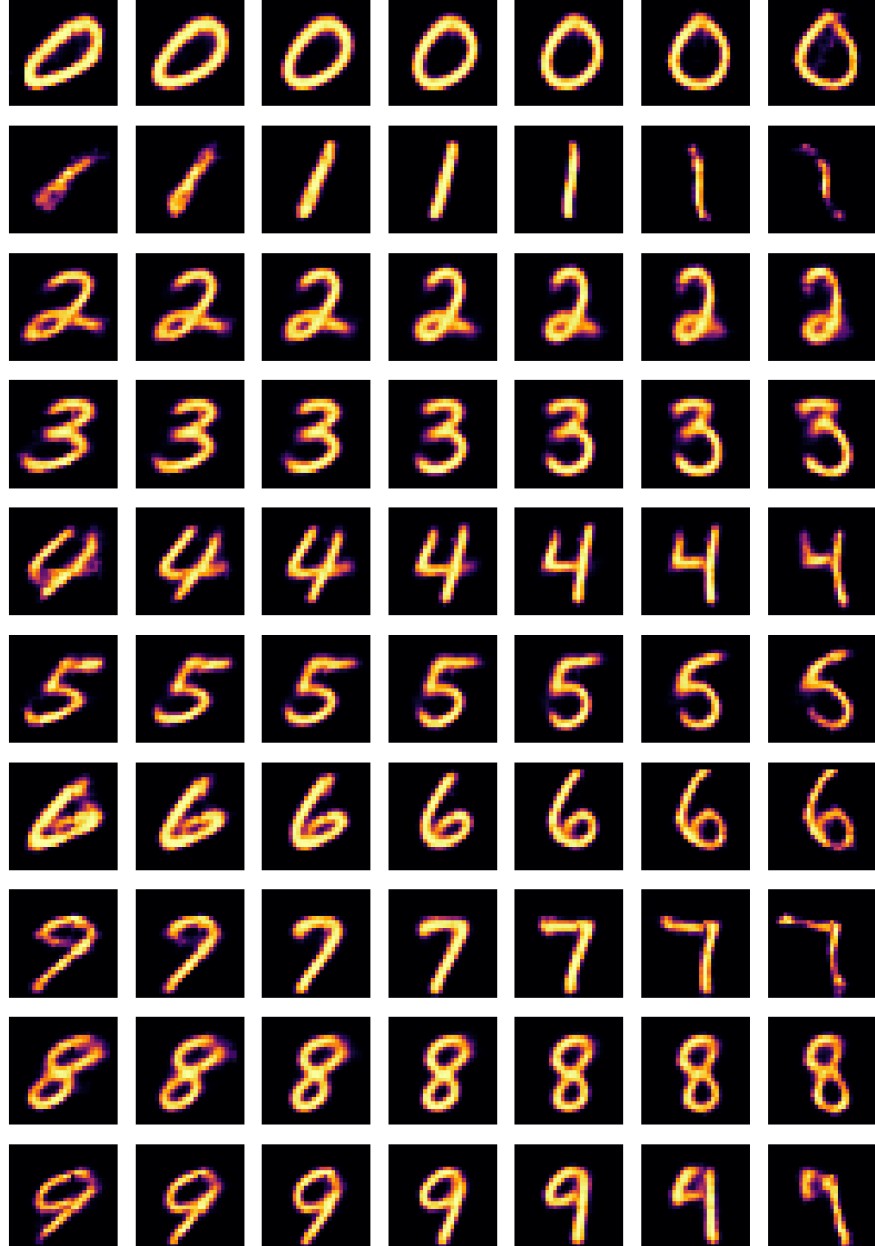

**Figure 8.** Symmetric morphing of images along streamlines of the 16-dimensional latent flow found in Section 5.3. The images in the middle column represent the "platonic" digits in the dataset (the cluster centers for each of the ten classes). The remaining six images in each row are obtained by moving along or against the streamline passing through the respective platonic image.

## 6. Summary and Outlook

In this study, we investigated a fundamental question in data science that is commonly encountered in many fields: what is the symmetry of a labeled dataset, and how does one identify its group structure? For this purpose, we applied a deep learning method that models generic symmetry transformation and its corresponding generators with a fully connected neural network. We trained the network with a specially constructed loss function, ensuring the desired symmetry properties, namely (i) the transformed samples preserve the oracle output (invariance); (ii) the transformations are non-trivial (normalization); (iii) in case of multiple symmetries, the learned transformations must be distinct (orthogonality) and (iv) must form an algebra (closure).

We leverage the fact that a non-trivial symmetry induces a corresponding flow in the latent space learned by an autoencoder. We demonstrated the performance of our method on the standard MNIST digit dataset, in which the oracle is provided by the 10-dimensional vector of logits of a trained classifier. We found classes of symmetries that transform each image from the dataset into new synthetic images while conserving the values of the logits. We illustrated these transformations as lines of equal probability ("flows") in the reduced latent space (see Figures 4, 6 and 7). These results show that symmetries in the data can be successfully searched for and identified as interpretable non-trivial transformations in the equivalent latent space.

The proposed method uniquely combines several advantages with respect to existing approaches in the literature.

- The method is agnostic (in the sense that we do not require any advance knowledge of what symmetries can be expected in the data) and non-parametric (the symmetry generators are a priori unrestricted, and their specific form is learned only during training). In other words, rather than testing for symmetries from a predefined list of possibilities, the symmetries are extracted directly from data.
- The symmetries are found in a reduced-dimensionality latent space, where the simple Euclidean metric is capable of capturing the relevant structure in the data (see Section 4.1).
- The oracle is allowed to be high-dimensional (in the case of the MNIST digits example, it is a 10-dimensional logit vector).

Future work could extend this approach to a much broader range of data and symmetry types of interest with respect to both formal theory and applied data science. For example, the presence of the closure term (15) can be leveraged in the study of continuous symmetries described by Lie groups. While classical Lie groups can be defined in terms of a single polynomial invariant [53], exceptional Lie groups require at least two invariants. On the data science side, there are many potential uses whenever the latent representation of the data reveals interesting structural features that are imperceptible or obscured in terms of the original high-dimensional features.

**Author Contributions:** Conceptualization, A.R., K.T.M. and K.M.; methodology, A.R., R.T.F., K.T.M., K.M. and E.B.U.; software, A.R., R.T.F. and E.B.U.; validation, A.R., R.T.F. and E.B.U.; formal analysis, A.R.; investigation, A.R., R.T.F. and E.B.U.; resources, K.T.M. and K.M.; data curation, A.R., R.T.F. and E.B.U.; writing—original draft preparation, K.T.M. and K.M.; writing—review and editing, A.R., K.T.M. and K.M.; visualization, A.R., R.T.F. and E.B.U.; supervision, K.T.M. and K.M.; project administration, K.T.M. and K.M.; funding acquisition, K.T.M. and K.M. All authors have read and agreed to the published version of the manuscript.

**Funding:** This research was funded by US Department of Energy (grant number DE-SC0022148). The APC was funded by DE-SC0022148.

**Institutional Review Board Statement:** Not applicable.

**Data Availability Statement:** The MNIST handwritten digit dataset [55] used in this study is one of the most popular openly available datasets in machine learning. It is included in most machine learning libraries, such as SCIKIT-LEARN, KERAS, TENSORFLOW, etc.

**Conflicts of Interest:** The authors declare no conflict of interest. The funders had no role in the design of the study; in the collection, analyses, or interpretation of data; in the writing of the manuscript; or in the decision to publish the results.

## Abbreviations

The following abbreviations are used in this manuscript:

| | |
|---|---|
| GAN | Generative Adversarial Network |
| ML | Machine learning |
| MDPI | Multidisciplinary Digital Publishing Institute |
| MNIST | Modified National Institute of Standards and Technology |

| MSE | Mean Squared Error |
| NN | Neural Network |
| ReLU | Rectified Linear Unit |
| SM | Standard Model |

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
