# Peer review of "Oracle-Preserving Latent Flows"

_symmetry, doi:10.3390/sym15071352_

Round 1

Reviewer 1 Report

This is an interesting attempt to model symmetry transformations and their corresponding generators by employing fully connected neural networks. These networks are trained using an ADAM optimizer and a specifically designed loss function that ensures the desired symmetry properties. The authors have involved a latent space with reduced dimensions and the extension of the methodology to encompass transformations. The contributions of the work are new and relevant to the journal's scope. The results are also well-attributed. There are a few minor and general concerns that need to be addressed as follows:

- The authors are suggested to include the critical findings (numerical values) in the Abstract. The references [8-14] and [23-35] in the Introduction section need to be explained further. The clubbing of such citations is unfair to the readers.

-  What are the limitations of existing state-of-the-art methods that motivate us to carry out this work?

- The authors are suggested including the algorithmic structure in Pytorch to implement the ANN approach to identify the symmetries.

Author Response

Response to Reviewer 1 Comments

Point 1: The authors are suggested to include the critical findings (numerical values) in the Abstract.

Response 1: We thank the referee for the feedback. Following the suggestions from both referees, we have revised the abstract in accordance with the journal style. 

Point 2: The references [8-14] and [23-35] in the Introduction section need to be explained further. The clubbing of such citations is unfair to the readers.

Response 2: We thank the referee for the suggestion. In the revised version we expanded the discussion of previous work and described those references individually.

Point 3: What are the limitations of existing state-of-the-art methods that motivate us to carry out this work?

Response 3: There are several advantages of our approach, which have been further emphasized in the revised version (introduction and conclusions).

  • the symmetries are found in the reduced dimensionality latent space, where the simple Euclidean metric is capable of capturing the relevant structure in the data (this was discussed in section 4.1)
  • the oracle is high dimensional (in this case 10-dimensional). This was mentioned in the abstract and section 2.
  • the method is completely general and agnostic (in the sense that it does not need to know what types of symmetries should be expected). Some of the previous related work in the literature was only focused on checking whether a certain given symmetry was realized in the data or not. This point was discussed in the last paragraph of the introduction and in the conclusions.

Point 4: The authors are suggested including the algorithmic structure in Pytorch to implement the ANN approach to identify the symmetries.

Response 4: We thank the referee for the suggestion. In addition to showing the ANN architecture in Figure 3, we have also posted our code publicly on github and in the revised version we included a link to the code.

Reviewer 2 Report

In my opinion, the structure of the article is appropriate. The purpose of the work was clearly defined. The scientific overtones are satisfying. The text and figures are very careful and are of good technical and substantive quality. Some final remarks:
1) In my opinion, the abstract lacks a conclusion. The abstract ends with the sentence: "The method is demonstrated with several examples on the MNIST digit dataset." But what's next? How did it end? What is the conclusion of the research? Please comment.

According to the Symmetry journal (Instructions for Authors) the abstract should be a single paragraph and should follow the style of structured abstracts, but without headings: a) Background: Place the question addressed in a broad context and highlight the purpose of the study; b) Methods: Describe briefly the main methods or treatments applied. Include any relevant preregistration numbers, and species and strains of any animals used; c) Results: Summarize the article's main findings; and d) Conclusion: Indicate the main conclusions or interpretations.

2) In conclusion, the authors wrote: "Future work could extend this approach to a much broader range of data and symmetry types." In my opinion, that's not enough. I am asking the authors to consider adding the following section in the article (for example, within section 6): Limitations and further research.

Author Response

Response to Reviewer 2 Comments

Point 1. In my opinion, the abstract lacks a conclusion. The abstract ends with the sentence: "The method is demonstrated with several examples on the MNIST digit dataset." But what's next? How did it end? What is the conclusion of the research? Please comment. According to the Symmetry journal (Instructions for Authors) the abstract should be a single paragraph and should follow the style of structured abstracts, but without headings: a) Background: Place the question addressed in a broad context and highlight the purpose of the study; b) Methods: Describe briefly the main methods or treatments applied. Include any relevant preregistration numbers, and species and strains of any animals used; c) Results: Summarize the article's main findings; and d) Conclusion: Indicate the main conclusions or interpretations.

Response 1: We thank the referee for the feedback. Following the suggestions from both referees, we have revised the abstract in accordance with the journal style.

Point 2. In conclusion, the authors wrote: "Future work could extend this approach to a much broader range of data and symmetry types." In my opinion, that's not enough. I am asking the authors to consider adding the following section in the article (for example, within section 6): Limitations and further research.

Response 2: We thank the referee for the feedback. In the revised version we changed the title of the last section from Summary and Conclusions to Summary and Outlook and added discussion on future directions.